# IMPROVING TEXT-TO-IMAGE GENERATION WITH INPUT-SIDE INFERENCE-TIME SCALING

## ABSTRACT

Recent advances in text-to-image (T2I) generation have achieved impressive results, yet existing models often struggle with simple or underspecified prompts, leading to suboptimal image-text alignment, aesthetics, and quality. We propose a prompt rewriting framework that leverages large language models (LLMs) to refine user inputs before feeding them into T2I backbones. Our approach introduces a carefully designed reward system and an iterative direct preference optimization (DPO) training pipeline, enabling the rewriter to enhance prompts without requiring supervised fine-tuning data. We evaluate our method across diverse T2I models and benchmarks. Results show that our prompt rewriter consistently improves image-text alignment, visual quality, and aesthetics, outperforming strong baselines. Furthermore, we demonstrate strong transferability by showing that a prompt rewriter trained on one T2I backbone generalizes effectively to others without needing to be retrained. We also systematically study scalability, evaluating how performance gains scale with the capacity of the large LLM used as the rewriter. These findings highlight that prompt rewriting is an effective, scalable, and practical model-agnostic strategy for improving T2I systems. We plan to release the code and trained prompt rewriters soon.

## 1 INTRODUCTION

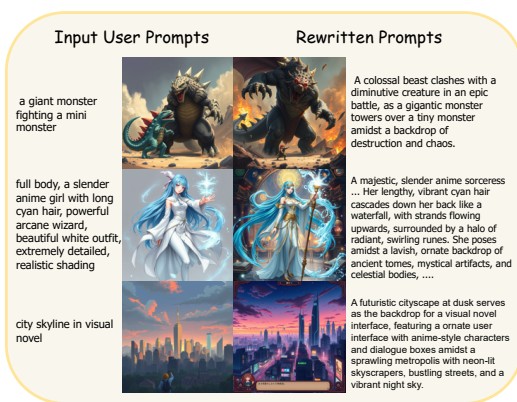

(a) Comparison between images generated from the original user prompts (left) and from rewritten, more detailed prompts (right), which yield results that are both more pleasing and better aligned with the input.

(b) Illustration of how rewritten prompts and their corresponding images progressively become more detailed and aesthetically pleasing across successive rounds of DPO training.

Figure 1: Rewritten prompts lead to images that are more detailed, visually pleasing, and better aligned with the input text, with quality improving through successive rounds of DPO training.

Text-to-image (T2I) generation has progressed rapidly along several fronts, including diffusion models (Wu et al., 2025a; Labs, 2024; Esser et al., 2024a), autoregressive models (Chen et al., 2025a; Wang et al., 2024), and emerging paradigms such as next-scale prediction (Tian et al., 2024), all of which demonstrate strong performance in terms of image quality and fidelity. However, one

important direction remains relatively underexplored: a simple, *model-agnostic*, and *training-free* pipeline that systematically addresses common shortcomings, such as weak text–image alignment and inconsistent aesthetics, by modifying only the input text.

In this work, we study an input-side, inference-time strategy that treats the T2I model as a black box and refines the user prompt, leaving the model parameters unchanged. This direction has *two manifolds of importance*: (i) from a practical perspective, real-world prompts are often short, vague, or underspecified, and rewriting them into clearer and more complete instructions mitigates alignment errors and stabilizes stylistic consistency; (ii) from a methodological perspective, it is of great interest to systematically study the achievable gain from optimizing the inputs only.

Prior work refines prompts by using the In-Context Learning ability of LLMs to expand or restructure the input text (Wu et al., 2025a; Hu et al., 2024), and by employing human-in-the-loop systems (Feng et al., 2023; Brade et al., 2023) that iteratively adjust prompts with feedback from complex engineering pipelines and human evaluators. These methods are not fully automated and depend on substantial human effort. Other approaches (Betker et al., 2023; Datta et al., 2024) train dedicated rewriters with supervised fine-tuning (SFT) on curated pairs of short and refined prompts. However, because different T2I models encode different preferences for "good" prompts, collecting high-quality, model-specific annotations is costly, may not transfer well across backbones, and ties improvements to a particular target model and dataset.

Inspired by inference-time scaling in LLMs (Muennighoff et al., 2025; Wu et al., 2024a; Chen et al., 2024a), which improves outputs by allocating more compute at test time, we adopt a complementary strategy for T2I that scales *on the input side*. Because output-side scaling is hard to unify across different T2I generators, we instead optimize the prompt that feeds a *frozen* T2I model. Concretely, we train a prompt rewriter using reinforcement learning with iterative Direct Preference Optimization (DPO) (Rafailov et al., 2023). Starting from a short user-provided prompt, the rewriter generates a set of candidate refined prompts. Each candidate is then used to synthesize images with a frozen T2I model, which are subsequently evaluated by multimodal LLM judges according to a composite reward function that integrates (i) image quality, (ii) aesthetics, and (iii) text–image alignment. Pairwise preferences inferred from these reward scores are employed to update the rewriter iteratively via DPO. Empirical evaluations on real-world user prompts demonstrate that our approach attains state-of-the-art performance, while avoiding the substantial costs associated with SFT data collection and curation. We further summarize our contributions below:

- We introduce **input-side inference-time scaling** to enhance T2I model performance, which is both **model-agnostic** and **training-free** for T2I models. We develop an RL-trained prompt rewriter that optimizes only the input text, improving both diffusion and autoregressive T2I backbones *without* SFT pairs, parameter updates, or architectural access.

- We conduct a systematic study of **scalability** and **transferability**. Specifically, we investigate (i) how performance *scales* with rewriter capacity, i.e., LLM size, and (ii) how an RL-trained rewriter *transfers* across diverse T2I backbones, quantifying cross-model robustness without per-model adaptation.

- Through extensive experiments across multiple LLMs, T2I models, and benchmarks, we show that our method consistently improves T2I performance in terms of image quality, aesthetics, and text–image alignment, achieving state-of-the-art results.

## 2 RELATED WORK

Prompts play a pivotal role in determining the quality and fidelity of outputs in downstream tasks. Consequently, a growing body of research (Kong et al., 2024; Li et al., 2024; Wang et al., 2025) has investigated techniques for prompt refinement in natural language processing. In the context of text-to-image (T2I) generation, prompt refinement methods can be broadly categorized as follows. We provide a more thorough discussion about related work in Appendix B.

**Learning-free Methods**  A number of works (Feng et al., 2023; Brade et al., 2023) explore interactive refinement approaches that leverage human input to produce more detailed and effective prompts for image synthesis. Mañas et al. (2024) propose a backpropagation-free optimization strategy that automatically rewrites prompts to improve prompt–image consistency. Similarly, Chen et al. (2024b) present a method that exploits historical user interactions to rewrite prompts, thereby better aligning generated images with individual user preferences.

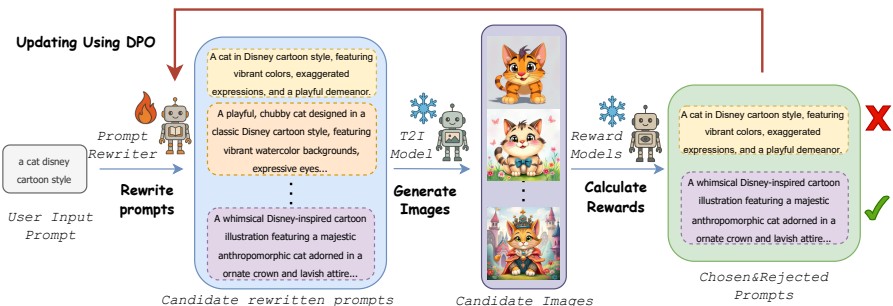

Figure 2: The Iterative DPO Training Pipeline. Given a user input prompt, the rewriter generates multiple candidate refinements, which are used by a frozen T2I model to synthesize images. Reward models evaluate these images to produce pairwise preferences. The chosen and rejected prompt pairs are then used to update the rewriter via Direct Preference Optimization, and the process is repeated iteratively for multiple rounds. The T2I model is kept frozen.

**Supervised Fine-Tuning** A straightforward strategy for prompt refinement is supervised fine-tuning. For instance, DALL-E 3 (Betker et al., 2023) employs an auxiliary image captioner trained to generate high-quality captions, which are subsequently used to recaption images and enhance T2I training. Datta et al. (2024) introduce a Prompt Expansion framework that automatically generates multiple aesthetically rich prompt variants from a single user query, thereby improving generation quality.

**Reinforcement Learning** Reinforcement learning (RL) has also been applied to prompt refinement. An early study by Hao et al. (2023) utilized PPO (Schulman et al., 2017) to fine-tune a GPT-2 model for prompt rewriting. Building on this direction, Wu et al. (2024b) combined SFT and PPO to mitigate prompt toxicity and generate safer images. Lee et al. (2024a) proposed a multi-reward RL framework that jointly fine-tunes a T2I diffusion model and a prompt-expansion network. A concurrent work, RePrompt (Wu et al., 2025b) explores RL-based refinement for T2I prompts. A detailed comparison between our method and RePrompt is provided in Appendix D.

## 3 METHOD

We train the rewriter with iterative DPO, without relying on SFT, and our results demonstrate that the rewriter can successfully learn the task purely through reinforcement learning. For each input prompt, the policy generates $n$ candidate rewrites, which are then passed to a frozen T2I model to synthesize images. A multimodal LLM judge evaluates these images and provides pairwise preferences across multiple criteria. The aggregated preferences yield a single winner–loser pair per prompt, which is used to optimize the DPO loss. This procedure is repeated over multiple rounds to progressively refine the rewriter. The overall pipeline is illustrated in Figure 2.

### 3.1 LEARNING ALGORITHM: ITERATIVE DPO WITHOUT SFT

Training a prompt rewriter for T2I models is challenging. SFT typically requires model-specific data curation, which is both costly and prone to biases tied to a particular backbone, thereby hindering transferability. Moreover, our preliminary experiments revealed that SFT diminishes the policy's exploratory capacity, resulting in overfitting to the training prompts rather than exhibiting robust generalization at test time. To overcome these limitations, we bypass SFT entirely and instead train the rewriter exclusively through iterative Direct Preference Optimization (DPO).

Concretely, at each training round the rewriter proposes multiple candidate rewrites for a user prompt $x$; images synthesized from these rewrites by a frozen T2I model are scored to produce pairwise preferences. We then update the policy via DPO using the following objective:

$$\mathcal{L}_{\text{DPO}}(\pi_\theta; \pi_{\text{ref}}) = -\mathbb{E}_{(x,y_w,y_l)\sim\mathcal{D}} \left[ \log \sigma \left( \beta \log \frac{\pi_\theta(y_w \mid x)}{\pi_{\text{ref}}(y_w \mid x)} - \beta \log \frac{\pi_\theta(y_l \mid x)}{\pi_{\text{ref}}(y_l \mid x)} \right) \right], \quad (1)$$

where $x$ is the input prompt, $y_w$ and $y_l$ represent the chosen and rejected rewritten prompts, respectively, and $\pi_{\text{ref}}$ is the reference model distribution.

We do not adopt GRPO (Shao et al., 2024; Guo et al., 2025a) or its variants, as these methods work best when a reliable scalar reward is available. In our setting, pointwise image rewards are too noisy to trust, so we rely on *pairwise* image comparisons to reduce variance. These pairwise signals map poorly to GRPO but align natively with DPO. Moreover, Tong et al. (2025) report that, in image generation settings with chain-of-thought reasoning, DPO yields stronger in-domain performance than GRPO, an empirical finding that aligns with our setting and supports our choice of DPO over GRPO.

## 3.2 Reward Design

We adopt the *MLLM-as-a-judge* paradigm and employ Qwen2.5-VL-72B-Instruct as the evaluation model. During training, the judge is provided with an instruction together with *two* candidate images, and it outputs a pairwise preference or a tie.

To capture the primary aspects of user interest, we define four reward dimensions: (1) image quality, $r_{\text{Quality}}$; (2) general image–text alignment, $r_{\text{General-Alignment}}$; (3) physical image–text alignment, $r_{\text{Physical-Alignment}}$; and (4) aesthetics, $r_{\text{Aesthetics}}$.

**Image Quality.** The image quality reward evaluates whether the generated image is plausible across five dimensions: (1) correctness of human or animal body parts, (2) plausibility of geometric details (e.g., absence of unintended wavy or misaligned lines, and consistency in symmetrical objects or features), (3) adherence to basic physical laws such as reflections and shadows, (4) semantic and logical coherence of the scene, and (5) correctness of fine-grained details without noticeable noise or distortions.

**General Image–Text Alignment.** This reward evaluates the degree to which an image adheres to the input prompt at a holistic level. Since assessing the entire prompt directly can be challenging, we employ a two-step procedure:

*Step 1. Decomposition.* Given the input prompt, the judge generates a concise list of key questions that capture its essential components. For instance, for the prompt "four apples on a table," the decomposition may yield: "Are there four apples?"; "Is there a table?"; and "Are the four apples on the table rather than below it?"

*Step 2. Comparison.* The decomposed questions, together with the candidate image pair, are then provided to the judge, which produces a pairwise preference decision.

**Physical Image–Text Alignment.** This reward is designed to assess concrete physical relationships that text-to-image models frequently fail to capture. Specifically, it evaluates: (i) the correctness of object counts, (ii) spatial relations such as left/right, on/under, and in front/behind, and (iii) attribute binding, i.e., whether the specified color, size, or style is accurately associated with the corresponding object.

**Image Aesthetics.** The aesthetics reward assesses the relative visual appeal and stylistic quality of two candidate images, taking into account factors such as composition, richness of detail, and overall artistic merit.

Based on these rewards, we train two distinct types of prompt rewriters: a *general rewriter*, which prioritizes semantic faithfulness and overall image quality, and an *aesthetics rewriter*, which emphasizes visual appeal while maintaining alignment and quality. We trained the two rewriters by combining the above rewards in different ways:

$$r^{\text{General Rewriter}} = r_{\text{Quality}} + r_{\text{General-Alignment}} + r_{\text{Physical-Alignment}}. \tag{2}$$

$$r^{\text{Aesthetics Rewriter}} = r_{\text{Quality}} + r_{\text{General-Alignment}} + r_{\text{Physical-Alignment}} + r_{\text{Aesthetics}}. \tag{3}$$

More details on the prompt design for the MLLM-as-a-judge reward are provided in Appendix H.

## 3.3 Selection of Chosen and Rejected Prompts

For each original prompt, we construct a single chosen-rejected pair for DPO training using pairwise image comparisons, which provide more reliable supervision than pointwise scoring.

Given a prompt $x$, we sample $n$ rewritten candidates and render $n$ corresponding images. From these, we form all $n(n-1)$ *ordered* pairs. Each ordered pair is evaluated by all reward models

associated with the current rewriter. For each comparison, the reward model issues a preference: the preferred image receives $+1$, the other receives $-1$, and a tie assigns $0$ to both.

We then aggregate these votes across all reward models and all ordered pairs to obtain a cumulative score for each rewritten candidate. The candidate with the highest score is selected as the chosen prompt $y_w$, while the one with the lowest score is designated as the rejected prompt $y_l$. The triplet $(x, y_w, y_l)$ is subsequently used as a DPO training instance. The complete procedure for selecting the chosen and rejected prompts is summarized in Alg. 1.

---

**Algorithm 1** Selection of chosen and rejected rewritten prompts

---

**Require:** Original prompt $x$, rewritten candidates $\{y_1, y_2, \ldots, y_n\}$, text-to-image model $g$, reward models $\{\text{RM}_k\}_{k=1}^{K}$
   **for** $i = 1$ to $n$ **do**
      Generate image $\text{Img}_i \leftarrow g(y_i)$
      Initialize rewards $r_i^k \leftarrow 0$ for all $k = 1, \ldots, K$
   **end for**
   **for** $i = 1$ to $n$ **do**
      **for** $j = 1$ to $n$ **such that** $j \neq i$ **do**
         **for** $k = 1$ to $K$ **do**
            **if** $\text{Img}_i \succ \text{Img}_j \mid \text{RM}_k, x$ **then**
               $r_i^k \leftarrow r_i^k + 1, \quad r_j^k \leftarrow r_j^k - 1$
            **else if** $\text{Img}_i \prec \text{Img}_j \mid \text{RM}_k, x$ **then**
               $r_i^k \leftarrow r_i^k - 1, \quad r_j^k \leftarrow r_j^k + 1$
            **end if**                          ▷ No update if $\text{Img}_i \approx \text{Img}_j \mid \text{RM}_k, x$
         **end for**
      **end for**
      Aggregate rewards: $R_i \leftarrow \sum_{k=1}^{K} r_i^k$
   **end for**
   Select chosen prompt $y_w \leftarrow y_{i_w}$, $i_w \leftarrow \arg\max_i R_i$
   Select rejected prompt $y_l \leftarrow y_{i_l}$, $i_l \leftarrow \arg\min_i R_i$
   **return** $(y_w, y_l)$

---

**Algorithm 2** Iterative DPO for Prompt Rewriting

---

**Require:** Base LLM $f_\theta$, training prompt set $\mathcal{D}$, samples-per-round $m$, candidates-per-prompt $n$
   **for** each round **do**
      Update reference policy $f_{ref} = f_\theta$.
      Sample prompts for each round: $\{x_i\}_{i=1}^{m} \sim \mathcal{D}$.
      **for** $i = 1$ to $m$ **do**
         Generate rewritten prompts: $\{y_i^j\}_{j=1}^{n} \sim f_\theta(\cdot \mid x_i)$.
         Get chosen and rejected rewritten prompts $(y_i^w, y_i^l)$ with Alg. 1.
      **end for**
      **for** each step **do**
         Sample a batch $B$ from $\{(x_i, y_i^w, y_i^l)\}_{i=1}^{m}$,
         Update $f_\theta$ with $f_{\text{ref}}$ and $B$ using Eq. 1.
      **end for**
   **end for**
   **return** $f_\theta$

---

### 3.4 ITERATIVE DPO PIPELINE

At each iteration of the DPO training, we draw a batch of prompts and employ the current rewriter to generate $n$ candidate rewrites for each prompt. Subsequently, Algorithm 1 is applied to the pairwise comparisons in order to identify a single preferred and a single rejected rewrite per prompt. The policy is then updated by minimizing the DPO loss defined in Equation 1. The overall training process is summarized in Algorithm 2.

Table 1: Pick-a-Pic v2 results with Llama-3-70B-Instruct as the LLM backbone. Reported values are GPT-4o–judged win rates vs. DALL·E 3. The general rewriter achieves the highest text–image alignment while also improving quality and aesthetics; the aesthetics rewriter delivers the best aesthetics while maintaining quality and alignment. **Bold** is the best and underline the second-best.

| | Image Quality | Image Aesthetics | Image-Text Alignment | Average | LAION |
|---|---|---|---|---|---|
| Dalle-3 | 0.500 | 0.500 | 0.500 | 0.500 | 6.19 |
| FLUX.1-schnell | 0.469 | 0.314 | 0.419 | 0.401 | 6.07 |
| + ICL | 0.475 | 0.307 | 0.422 | 0.401 | 6.07 |
| + Ours (General Rewriter) | 0.494 | 0.476 | **0.561** | 0.510 | 6.08 |
| + Ours (Aesthetics Rewriter) | **0.495** | **0.818** | 0.424 | **0.579** | **6.39** |
| FLUX.1-dev | 0.513 | 0.350 | 0.360 | 0.408 | 6.22 |
| + ICL | 0.531 | 0.363 | 0.457 | 0.450 | 6.17 |
| + Ours (General Rewriter) | 0.536 | 0.491 | **0.575** | 0.534 | 6.06 |
| + Ours (Aesthetics Rewriter) | **0.576** | **0.800** | 0.391 | **0.589** | **6.41** |
| SD-3.5-medium | 0.424 | 0.230 | 0.425 | 0.359 | 5.74 |
| + ICL | 0.487 | 0.327 | 0.455 | 0.423 | 5.92 |
| + Ours (General Rewriter) | **0.504** | 0.456 | **0.542** | 0.501 | 5.95 |
| + Ours (Aesthetics Rewriter) | 0.481 | **0.799** | 0.401 | **0.560** | **6.23** |
| JanusPro | 0.193 | 0.181 | 0.392 | 0.255 | 5.89 |
| + ICL | **0.213** | 0.180 | 0.335 | 0.243 | 5.97 |
| + Ours (General Rewriter) | 0.183 | 0.268 | **0.421** | 0.291 | 5.95 |
| + Ours (Aesthetics Rewriter) | 0.200 | **0.612** | 0.317 | **0.377** | **6.23** |

## 4 EXPERIMENT

We provide the details of LLMs for prompt rewriter, T2I models, training datasets, baselines, as well as the detailed experimental settings and hyperparameters in Appendix C.

### 4.1 MLLM-AS-A-JUDGE RESULTS

On Pick-a-Pic v2, as reported in Table 1, and with GPT-4o serving as the evaluation judge, our rewriters consistently outperform the original test prompts without requiring additional training of the underlying T2I backbones. The general rewriter attains the highest win rate on text–image alignment while also improving image quality and aesthetics. The aesthetics rewriter achieves the strongest performance in terms of visual appeal, with an aesthetics win rate of 0.818, while maintaining competitive results on both image quality and alignment.

On PartiPrompts, as reported in Appendix Table 7, GPT-4o evaluation demonstrates consistent improvements across all T2I backbones when combined with our rewriter, with win rates compared to original user input prompts exceeding 0.5 for every evaluation dimension.

We further observe that simple in-context learning (ICL) prompts frequently improve generation quality. Contemporary T2I models often underperform when conditioned on short or underspecified prompts, which are prevalent in real-world usage, whereas straightforward prompt expansions provide consistent benefits. This phenomenon likely arises from a distributional gap, as training captions are generally longer and more descriptive than prompts provided by real users.

### 4.2 BENCHMARK RESULTS

As shown in Tables 2 and 3, our method achieves strong performance on GenEval, T2I-CompBench++, and the TIFA-Benchmark in terms of text–image alignment, and attains lower FID scores on MS-COCO-30K for image quality. Across all benchmarks, our approach consistently improves alignment, for example increasing the FLUX.1-dev GenEval score from 0.70 to 0.79, while simultaneously enhancing image quality as reflected by reduced FID.

### 4.3 BASELINES COMPARISON

We compile recent state-of-the-art baseline methods that report on GenEval, spanning both autoregressive (Show-o) and diffusion (FLUX) T2I families. Under the same evaluation protocol (Table 4),

Table 2: GenEval results for the general rewriter across T2I backbones. It consistently outperforms the original prompts and ICL baselines, and performance improves with LLM size; Llama-3-70B-Instruct achieves the best results.

| Model | Single object | Two object | Counting | Colors | Position | Color attribution | Overall |
|---|---|---|---|---|---|---|---|
| FLUX.1-schnell | 0.99 | 0.87 | 0.61 | 0.79 | 0.35 | 0.43 | 0.67 |
| + ICL | 0.99 | 0.87 | 0.55 | 0.81 | 0.50 | 0.50 | 0.69 |
| + Ours (Llama 3 3B) | **1.00** | 0.89 | 0.63 | 0.81 | 0.47 | **0.59** | 0.73 |
| + Ours (Llama 3 8B) | **1.00** | **0.93** | 0.58 | **0.86** | 0.55 | 0.55 | 0.74 |
| + Ours (Llama 3 70B) | 0.99 | 0.91 | **0.65** | 0.77 | **0.64** | 0.57 | **0.75** |
| FLUX.1-dev | **1.00** | 0.87 | 0.76 | 0.84 | 0.22 | 0.49 | 0.70 |
| + ICL | **1.00** | 0.89 | **0.80** | 0.83 | 0.54 | 0.41 | 0.74 |
| + Ours (Llama 3 70B) | **1.00** | **0.95** | 0.78 | **0.88** | **0.58** | **0.56** | **0.79** |
| SD-3.5-medium | **1.00** | 0.84 | **0.68** | 0.85 | 0.24 | **0.61** | 0.70 |
| + ICL | 0.96 | **0.92** | 0.60 | **0.86** | 0.42 | 0.54 | 0.72 |
| + Ours (Llama 3 70B) | **1.00** | **0.92** | **0.68** | 0.85 | **0.57** | 0.56 | **0.76** |

Table 3: Results on T2I-CompBench++, TIFA-Benchmark, and FID on MS COCO 30K using the general rewriter with Llama-3-70B-Instruct as the LLM backbone. The rewriter improves text–image alignment and image quality across all evaluated T2I backbones.

| | T2I-CompBench++ | | | | | | TIFA | COCO |
|---|---|---|---|---|---|---|---|---|
| | Color↑ | Shape↑ | Texture↑ | Spatial↑ | Numeracy↑ | Complex↑ | Score↑ | FID↓ |
| FLUX.1-schnell | 0.7492 | 0.5673 | 0.6911 | 0.2754 | 0.6062 | 0.3614 | 0.8803 | 20.57 |
| + Ours | **0.7614** | **0.6010** | **0.7063** | **0.3216** | **0.6161** | **0.3713** | **0.8868** | **17.76** |
| FLUX.1-dev | 0.7647 | 0.5053 | 0.6336 | 0.2763 | 0.6130 | 0.3586 | 0.8572 | 24.38 |
| + Ours | **0.7978** | **0.5834** | **0.6929** | **0.3206** | **0.6343** | **0.3694** | **0.8809** | **19.57** |
| SD-3.5-medium | 0.7988 | 0.5800 | 0.7206 | 0.2889 | 0.6033 | 0.3614 | 0.8782 | 17.81 |
| + Ours | **0.8040** | **0.5855** | **0.7346** | **0.3322** | **0.6320** | **0.3707** | **0.8878** | **17.13** |
| JanusPro | 0.5294 | 0.3247 | 0.4168 | 0.1579 | 0.4380 | 0.3778 | 0.8457 | 19.28 |
| + Ours | **0.7861** | **0.5892** | **0.7220** | **0.2773** | **0.5983** | **0.3911** | **0.8845** | **16.71** |

our method achieves the best overall GenEval score, outperforming all other methods across both architecture types by a substantial margin.

## 4.4 Case Study

We provide qualitative examples in Appendix G that trace the evolution of rewritten prompts throughout the iterative DPO training process. Additionally, we visualize the stylistic differences between the outputs of the general rewriter, which prioritizes faithfulness, and the aesthetics rewriter, which focuses on visual appeal. Detailed examples can be found in Appendix Table 16.

## 5 Ablation Study

### 5.1 Reward Type

Table 10 presents an ablation study of the reward terms. Removing the quality reward substantially decreases the image–quality win rate, from 0.494 to 0.364. Excluding either the general or physical alignment rewards reduces alignment, from 0.561 to 0.521 and to 0.504 respectively, confirming that each reward effectively drives its intended objective. Incorporating the aesthetics reward yields a pronounced improvement in aesthetics, increasing from 0.476 to 0.818, but simultaneously reduces alignment from 0.561 to 0.424, highlighting a trade-off. Qualitative analysis in Table 16 suggests that the aesthetics reward often enriches scenes by introducing additional objects or ornamentation, which can dilute the main subject and thereby weaken alignment. To accommodate different objectives, we report two variants: a general rewriter optimized for faithfulness and overall quality, and an aesthetics rewriter tailored to visual appeal.

Table 4: GenEval results compared with state-of-the-art methods. Using the general rewriter with Llama-3-70B-Instruct as the LLM backbone, our method achieves the best overall GenEval score (higher is better). †Results reported by Wu et al. (2025b). ‡Results reported by Guo et al. (2025b).

| Method | T2I Backbone | Single object | Two object | Counting | Colors | Position | Color attribution | Overall |
|---|---|---|---|---|---|---|---|---|
| Baseline‡ | Show-o | 0.95 | 0.52 | 0.49 | 0.82 | 0.11 | 0.28 | 0.53 |
| Baseline | FLUX.1-dev | 1.00 | 0.87 | 0.76 | 0.84 | 0.22 | 0.49 | 0.70 |
| PARM‡ | Show-o | 0.99 | 0.77 | 0.68 | 0.86 | 0.29 | 0.45 | 0.67 |
| PARM++‡ | Show-o | 0.99 | 0.71 | 0.69 | **0.95** | 0.36 | 0.49 | 0.70 |
| Idea2Img† | FLUX.1-dev | - | - | - | - | - | - | 0.69 |
| Zero-Shot | FLUX.1-dev | 0.98 | 0.89 | 0.71 | 0.82 | 0.47 | 0.49 | 0.73 |
| ICL | FLUX.1-dev | **1.00** | 0.89 | **0.80** | 0.83 | 0.54 | 0.41 | 0.74 |
| RePrompt† | FLUX.1-dev | 0.98 | 0.87 | 0.77 | 0.85 | **0.62** | 0.49 | 0.76 |
| Ours | FLUX.1-dev | **1.00** | **0.95** | 0.78 | 0.88 | 0.58 | **0.56** | **0.79** |

## 5.2 Rewritten Prompt Length

Figure 5 illustrates a steady increase in the average token length of rewritten prompts over training. The rewriter progressively incorporates additional specificity, such as attributes, relations, and constraints, and this increase in length correlates with higher win rates.

## 5.3 Full Finetune vs. LoRA

We conduct a controlled comparison between LoRA and full-model finetuning under identical experimental conditions, as shown in Figure 6. Full-model finetuning does not yield consistent improvements, whereas LoRA attains comparable performance while requiring substantially less memory and computation. These results suggest that LoRA constitutes a strong and practical choice for prompt-rewriter training.

## 6 Analyses

### 6.1 Scalability

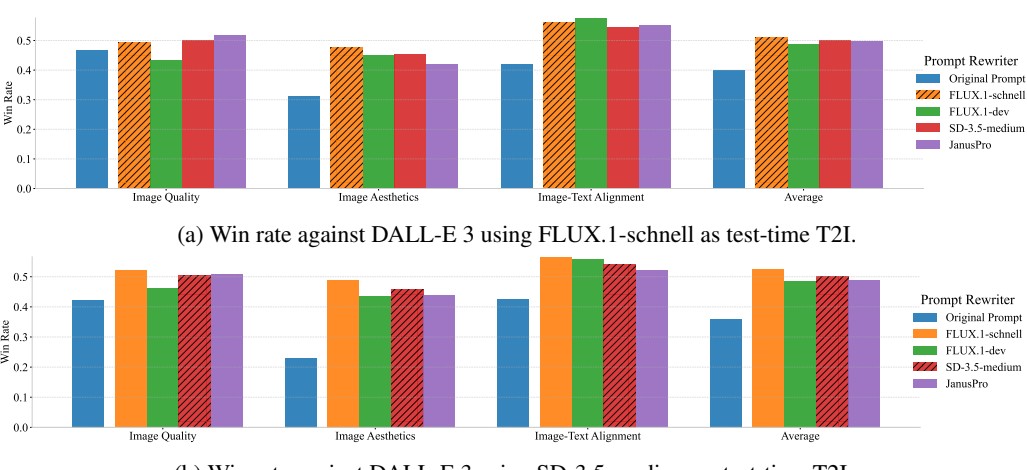

(a) Win rate against DALL-E 3 using FLUX.1-schnell as test-time T2I.

(b) Win rate against DALL-E 3 using SD-3.5-medium as test-time T2I.

Figure 3: Win rates for rewriters trained on various T2I backbones but evaluated on a fixed target, either (a) FLUX.1-schnell or (b) SD-3.5-medium. Evaluation is on the Pick-a-pic v2 dataset using a GPT-4o judge. Results show strong transferability.

Table 5: GPT-4o judged win rates vs. DALL-E 3 on the Pick-a-Pic v2 dataset when using various LLMs as the rewriter backbone. The results demonstrate a clear scaling trend: rewriter performance improves with the size of the LLM, with Llama-3-70B achieving the highest average win rate.

| | Image Quality | Image Aesthetics | Image-Text Alignment | Average |
|---|---|---|---|---|
| FLUX.1-schnell | 0.469 | 0.314 | 0.419 | 0.401 |
| + Ours (Qwen2.5 14B) | 0.455 | 0.275 | 0.452 | 0.394 |
| + Ours (Qwen2.5 32B) | 0.454 | 0.277 | 0.431 | 0.387 |
| + Ours (Qwen2.5 72B) | **0.458** | **0.305** | **0.465** | **0.409** |
| + Ours (Qwen3 8B) | 0.511 | 0.376 | 0.491 | 0.459 |
| + Ours (Qwen3 14B) | 0.490 | 0.362 | 0.473 | 0.442 |
| + Ours (Qwen3 32B) | **0.517** | **0.424** | **0.534** | **0.492** |
| + Ours (DeepSeek 8B) | **0.527** | 0.402 | 0.461 | 0.463 |
| + Ours (DeepSeek 32B) | 0.499 | **0.405** | 0.503 | **0.469** |
| + Ours (DeepSeek 70B) | 0.502 | 0.380 | **0.506** | 0.463 |
| + Ours (Llama3 3B) | **0.517** | 0.455 | 0.524 | 0.499 |
| + Ours (Llama3 8B) | 0.511 | 0.441 | **0.565** | 0.506 |
| + Ours (Llama3 70B) | 0.494 | **0.476** | 0.561 | **0.510** |

We systematically vary the rewriter's backbone across four LLM families and sizes to quantify input-side scaling (Table 5). Larger backbones generally yield higher averaged GPT-4o win rates, with the strongest average obtained using Llama-3-70B-Instruct.

Improvements are not uniform across metrics: smaller models can occasionally lead on a single axis (e.g., alignment or aesthetics), reflecting inherent trade-offs in prompt rewriting. This pattern is consistent with the reward ablation study above, where emphasizing aesthetics increases aesthetic preference but reduces alignment (Table 10).

Models tuned for explicit reasoning (e.g., DeepSeek-R1-Distilled series) underperform Llama-3-70B-Instruct overall, especially on aesthetics, suggesting that reasoning-oriented tuning can conflict with aesthetic goals and may limit exploratory prompt enrichment during rollout.

## 6.2 TRANSFERABILITY

We evaluate transferability by training multiple rewriters, each paired with a different T2I backbone, and subsequently freezing the rewriter for testing on a fixed target backbone. We consider two target models, FLUX.1-schnell and Stable Diffusion 3.5. For each target, we report performance in three conditions: when the training and target T2I backbones are identical, when they differ, and when using the original unmodified prompt as a baseline. Results are presented in Figure 3 and Table 8.

Across both targets, rewritten prompts consistently improve over the original prompts, irrespective of whether the training and testing backbones match. Performance differences under train–test mismatch are generally small and often comparable to the matched setting. These findings suggest that the rewriter acquires prompt refinements that generalize across T2I backbones, enabling reuse without per-model retraining. We hypothesize that this transferability arises because T2I models are commonly trained on image–caption pairs from similar distributions, rendering our rewrites broadly portable.

## 7 CONCLUSION

In this work, we propose an inference-time scaling framework that enhances text-to-image generation through LLM-based prompt rewriting. Our approach, trained with iterative DPO and composite rewards, consistently improves image quality, alignment, and aesthetics across diverse T2I backbones without modifying or retraining them. Experiments further demonstrate scalability, transferability, and a controllable trade-off between faithfulness and visual appeal, establishing input-side optimization as a practical and versatile strategy for advancing T2I systems.

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

## A USE OF LLMS

LLMs are employed to refine our writing and to provide assistance with coding.

## B RELATED WORK

### B.1 IN-CONTEXT LEARNING

In-Context Learning (ICL) has demonstrated strong effectiveness across a wide range of text-related tasks. Extending this concept to text-to-image generation, Zeng et al. (2024) formally define T2I ICL and introduce a benchmark for evaluating the in-context capabilities of T2I models. Several recent models, including Image-Qwen (Wu et al., 2025a), Emu2 (Sun et al., 2024), and ELLA (Hu et al., 2024), report performance gains attributable to ICL. Furthermore, Lee et al. (2024b) apply in-context few-shot learning strategies to further enhance T2I generation quality.

### B.2 CHAIN-OF-THOUGHT REASONING

Chain-of-thought (CoT) reasoning has proven highly effective for complex text-based tasks. Guo et al. (2025b) provide the first systematic study on adapting CoT techniques to autoregressive image generation, introducing specialized reward models that iteratively verify and refine outputs during the generation process. Complementarily, ImageGen-CoT (Liao et al., 2025) proposes an automated pipeline for constructing high-quality datasets to fine-tune unified multimodal large language models, thereby enhancing their contextual reasoning abilities and improving image generation quality.

### B.3 PROMPT REFINEMENT

Prompts play a pivotal role in determining the quality and fidelity of outputs in downstream tasks. Consequently, a growing body of research (Kong et al., 2024; Li et al., 2024; Wang et al., 2025) has investigated techniques for prompt refinement in natural language processing. In the context of text-to-image (T2I) generation, prompt refinement methods can be broadly categorized as follows.

**Learning-free Methods** A number of works (Feng et al., 2023; Brade et al., 2023) explore interactive refinement approaches that leverage human input to produce more detailed and effective prompts for image synthesis. Mañas et al. (2024) propose a backpropagation-free optimization strategy that automatically rewrites prompts to improve prompt–image consistency. Similarly, Chen et al. (2024b) present a method that exploits historical user interactions to rewrite prompts, thereby better aligning generated images with individual user preferences.

**Supervised Fine-Tuning** A straightforward strategy for prompt refinement is supervised fine-tuning. For instance, DALL-E 3 (Betker et al., 2023) employs an auxiliary image captioner trained to generate high-quality captions, which are subsequently used to recaption images and enhance T2I training. Datta et al. (2024) introduce a Prompt Expansion framework that automatically generates multiple aesthetically rich prompt variants from a single user query, thereby improving generation quality.

**Reinforcement Learning** Reinforcement learning (RL) has also been applied to prompt refinement. An early study by Hao et al. (2023) utilized PPO (Schulman et al., 2017) to fine-tune a GPT-2 model for prompt rewriting. Building on this direction, Wu et al. (2024b) combined SFT and PPO to mitigate prompt toxicity and generate safer images. Lee et al. (2024a) proposed a multi-reward RL framework that jointly fine-tunes a T2I diffusion model and a prompt-expansion network. A concurrent work, RePrompt (Wu et al., 2025b) explores RL-based refinement for T2I prompts. A detailed comparison between our method and RePrompt is provided in Appendix D.

## C EXPERIMENT SETTING DETAILS

### C.1 MODELS AND DATASETS

We conduct training on two datasets: Pick-a-Pic v2 (Kirstain et al., 2023), which consists of real user prompts, and T2I-CompBench++ (Huang et al., 2025), which provides compositional prompts designed to evaluate text–image alignment. For T2I-CompBench++, we adopt the official training split. Since Pick-a-Pic v2 does not provide predefined splits, we partition the data ourselves and utilize only the training portion. In total, our training corpus contains approximately 60k prompts. For the Pick-a-Pic v2 test split, we keep 500 prompts for evaluations. As our approach does not require SFT, only the prompt text is needed, making the data curation process text-only and considerably simpler than collecting paired text–image data.

To validate universal applicability of our method, we experiment on four families of large language models: Llama (3.3-70B-Instruct, 3.1-8B-Instruct, 3.2-3B-Instruct) (Dubey et al., 2024), DeepSeek-R1-Distill (70B/32B/8B) (Guo et al., 2025a), Qwen3 (32B/14B/7B) (Yang et al., 2025), and Qwen2.5 (72B/32B/14B) (Team, 2024), as well as four representative text-to-image generation backbones: Flux.1-Schnell and Flux.1-Dev (Labs, 2024), Stable Diffusion 3.5 Medium (Esser et al., 2024b), and JanusPro (Chen et al., 2025b).

### C.2 EVALUATION METRICS AND BENCHMARKS

We employ both standard benchmarks and multimodal LLM-based judgment. Image quality is assessed using the Fréchet Inception Distance (FID) on MS-COCO-30K (Lin et al., 2014), following the protocol of Pavlov et al. (2023). Text–image alignment is evaluated on GenEval (Ghosh et al., 2023), the T2I-CompBench++ (Huang et al., 2025) test split, and the TIFA Benchmark (Hu et al., 2023). Aesthetics is measured with a LAION-based predictor (Schuhmann et al., 2022). To complement these automated metrics, we further employ GPT-4o as a multimodal LLM-as-a-judge on the Pick-a-Pic v2 (Kirstain et al., 2023) test split and PartiPrompts (Yu et al., 2022), where it provides pairwise ratings for image quality, text–image alignment, and aesthetics.

### C.3 SETTINGS AND HYPERPARAMETERS

The models were trained on 64 NVIDIA H100-SXM GPUs, with the training parameters summarized in Table 6. For image generation with T2I models, a resolution of $1024 \times 1024$ was used for FLUX.1-schnell, FLUX.1-dev, and SD-3.5-medium, while a resolution of $384 \times 384$ was used for the JanusPro experiments. The default generation parameters for T2I were applied.

### C.4 BASELINES

We compare our method with multiple baselines:

- **In-context learning (ICL)**: We prompt the model with a set of curated examples and detailed prompts to rewrite the model.
- **Idea2Img** (Yang et al., 2023) employs GPT-4V to iteratively refine prompts through feedback, memory, and draft image selection, thereby enhancing automatic image design.
- **PARM** (Guo et al., 2025b) performs adaptive step-wise evaluation. It introduces three tasks: (1) clarity judgment to skip blurry early steps, (2) potential assessment to prune low-potential paths once images are sufficiently clear, and (3) best-of-N selection across high-potential candidates
- **PARM++** (Guo et al., 2025b) extends PARM with a reflection mechanism. After candidate selection, it performs reflection evaluation to check for image-text misalignments. If discrepancies are detected (e.g., wrong objects, colors, or positions), PARM++ generates explicit reasons and instructs the generative model to iteratively self-correct its outputs until alignment is achieved.
- **Reprompt** (Wu et al., 2025b) trains a language model to produce structured, reasoning-driven prompts via GRPO. We discuss the detailed differences between Reprompt and our work in Sec. D.

Table 6: Training parameters in our experiments.

| Parameter Name | Value |
|---|---|
| Prompt Rewriter temperature | 1.5 |
| DPO $\beta$ | 0.1 |
| maximum iterative DPO round | 6 |
| samples-per-round $m$ | 10000 |
| candidates-per-prompt $n$ | 5 |
| epochs per DPO round | 5 |
| lora rank | 64 |
| DPO batch size | 256 |
| learning rate | 5.0e-6 |
| lr scheduler | cosine |
| warmup ratio | 0.1 |
| optimizer | AdamW |
| AdamW $\beta$ | (0.9, 0.999) |
| AdamW $\epsilon$ | 1e-8 |
| AdamW weight decay | 0.01 |

## D  DIFFERENCES WITH REPROMPT

In this section, we provide a detailed comparison between our method and the concurrent work Reprompt (Wu et al., 2025b).

From the algorithmic perspective, our approach does not include an SFT stage. Our preliminary experiments revealed that SFT constrains the exploration ability of LLMs, leading to overfitting on the SFT training data. Moreover, collecting diverse, high-quality, and unbiased training data tailored to each T2I model is both challenging and costly. By directly applying RL for training, our method only requires raw user prompts as inputs, without the need for model-specific datasets.

Another key distinction lies in the optimization paradigm. Reprompt adopts GRPO, whereas we employ DPO. We argue that reward estimation for generated images is inherently difficult under GRPO, while the construction of chosen and rejected prompt pairs is relatively easy. This enables more reliable reward guidance in DPO. Consistent with our findings, Tong et al. (2025) also report that DPO achieves superior in-domain performance compared to GRPO.

In terms of reward design, Reprompt primarily focuses on text–image alignment and only reports results on GenEval and T2I-Compbench, both of which emphasize alignment evaluation. However, no results are provided for image quality or aesthetics. In contrast, our framework evaluates these additional dimensions by reporting image quality and aesthetics metrics on Pick-a-Pic v2 and PartiPrompts, as well as FID scores on MS COCO 30k. Our experiments further reveal a clear trade-off among alignment, quality, and aesthetics, highlighting the necessity of improving alignment without compromising other aspects.

From the experimental perspective, we comprehensively evaluated the scalability and transferability. Whereas Reprompt is limited to Qwen2.5-3B, we incorporate a broader range of LLMs and T2I model backbones, and also conduct large-scale experiments with 70B-parameter LLMs. These experiments demonstrate both data scalability and model scalability: as illustrated in Figure 4, performance improves with larger training datasets, and as shown in Table 5, larger LLMs further enhance results.

Finally, our method consistently outperforms Reprompt on GenEval and T2I-Compbench, while simultaneously advancing performance on image quality and aesthetics benchmarks.

# E ADDITIONAL EXPERIMENTS

## E.1 PARTIPROMPTS

We present additional results on PartiPrompts in Table 7. The reported win rates demonstrate that our method consistently enhances image quality, text–image alignment, and aesthetics.

Table 7: PartiPrompts (Yu et al., 2022) results. Values are GPT-4o–judged win rates (higher is better) of our rewritten prompts vs. the original prompts. Our method consistently improves image quality, alignment, and aesthetics.

| | Image Quality | Image Aesthetics | Image-Text Alignment | Average |
|---|---|---|---|---|
| FLUX.1-schnell | 0.524 | 0.583 | 0.592 | 0.566 |
| FLUX.1-dev | 0.510 | 0.584 | 0.625 | 0.573 |
| SD-3.5-medium | 0.599 | 0.743 | 0.542 | 0.628 |
| JanusPro | 0.616 | 0.674 | 0.687 | 0.659 |

## E.2 TRANSFERABILITY

We further evaluate the transferability on the GenEval datasets in Table 8. The results show that, even when the training and testing T2I backbones differ, our method still achieves good performance, demonstrating robust transferability.

## E.3 PERFORMANCE OF JANUSPRO ON GENEVAL

We report counterintuitive results for JanusPro on GenEval in Table 9, where both ICL and our method fail to yield improvements. In contrast, our approach achieves performance gains for Janus-Pro on T2I-CompBench++ and TIFA-Benchmark. Interestingly, JanusPro outperforms the other three diffusion-based models on GenEval, yet performs consistently worse on T2I-CompBench++ and TIFA-Benchmark. We hypothesize that this discrepancy arises from JanusPro potentially over-fitting to the GenEval benchmark.

Table 8: Transferability results evaluated on GenEval. We train rewriters on different T2I backbones and evaluate on fixed target backbones (FLUX.1-schnell and SD-3.5-medium). The results demonstrate that, even when the training and testing backbones differ, our method still achieves significant performance improvements. "()" shows what T2I model is used during training.

| Model | Single object | Two object | Counting | Colors | Position | Color attribution | Overall |
|---|---|---|---|---|---|---|---|
| FLUX.1-schnell w/ orig. prompt | 0.99 | 0.87 | 0.61 | 0.79 | 0.35 | 0.43 | 0.67 |
| w/ rewriter (FLUX.1-schnell) | 0.99 | 0.91 | 0.65 | 0.77 | **0.64** | **0.57** | 0.75 |
| w/ rewriter (FLUX.1-dev) | 0.99 | 0.93 | 0.66 | **0.86** | 0.55 | **0.57** | 0.76 |
| w/ rewriter (SD-3.5-medium) | **1.00** | **0.95** | **0.72** | 0.81 | 0.61 | 0.66 | **0.79** |
| w/ rewriter (JanusPro) | 0.99 | 0.87 | 0.63 | 0.81 | 0.56 | **0.57** | 0.74 |
| SD-3.5-medium w/ orig. prompt | **1.00** | 0.84 | 0.68 | 0.85 | 0.24 | 0.61 | 0.70 |
| w/ rewriter (FLUX.1-schnell) | **1.00** | **0.92** | **0.69** | 0.83 | **0.58** | **0.69** | **0.78** |
| w/ rewriter (FLUX.1-dev) | **1.00** | **0.92** | 0.68 | **0.88** | 0.51 | 0.60 | 0.76 |
| w/ rewriter (SD-3.5-medium) | **1.00** | **0.92** | 0.68 | 0.85 | 0.57 | 0.56 | 0.76 |
| w/ rewriter (JanusPro) | **1.00** | 0.88 | 0.70 | 0.86 | 0.54 | 0.54 | 0.75 |

Table 9: GenEval results for JanusPro, where both ICL and our method do not yield performance improvements.

| Model | Single object | Two object | Counting | Colors | Position | Color attribution | Overall |
|---|---|---|---|---|---|---|---|
| JanusPro | **1.00** | 0.86 | **0.59** | **0.89** | **0.73** | **0.65** | **0.79** |
| + ICL | **1.00** | **0.93** | 0.39 | 0.86 | 0.43 | 0.48 | 0.68 |
| + Ours | **1.00** | 0.81 | 0.38 | 0.86 | 0.60 | 0.43 | 0.68 |

Table 10: Reward ablation results for the prompt rewriter. Removing any individual reward leads to a decline in its associated metric. Incorporating the aesthetics reward improves aesthetic preference but decreases alignment, thereby revealing an inherent trade-off.

| | Rewriter Type | Image Quality | Image Aesthetics | Image-Text Alignment | Average |
|---|---|---|---|---|---|
| FLUX.1-schnell | - | 0.469 | 0.314 | 0.419 | 0.401 |
| + ICL | - | 0.475 | 0.307 | 0.422 | 0.401 |
| + Zero Shot | - | 0.431 | 0.263 | 0.417 | 0.370 |
| + Ours | General Rewriter, w/o quality reward | 0.364 | 0.461 | 0.563 | 0.463 |
| + Ours | General Rewriter, w/o general alignment reward | 0.523 | 0.470 | 0.521 | 0.505 |
| + Ours | General Rewriter, w/o physical alignment reward | 0.533 | 0.448 | 0.504 | 0.495 |
| + Ours | General Rewriter | 0.494 | 0.476 | 0.561 | 0.510 |
| + Ours | Aesthetics Rewriter | 0.495 | 0.818 | 0.424 | 0.579 |

## F ABLATION STUDY

### F.1 REWARD TYPE

Table 10 presents an ablation study of the reward terms. Removing the quality reward substantially decreases the image–quality win rate, from 0.494 to 0.364. Excluding either the general or physical alignment rewards reduces alignment, from 0.561 to 0.521 and to 0.504 respectively, confirming that each reward effectively drives its intended objective. Incorporating the aesthetics reward yields a pronounced improvement in aesthetics, increasing from 0.476 to 0.818, but simultaneously reduces alignment from 0.561 to 0.424, highlighting a trade-off. Qualitative analysis in Table 16 suggests that the aesthetics reward often enriches scenes by introducing additional objects or ornamentation, which can dilute the main subject and thereby weaken alignment. To accommodate different objectives, we report two variants: a general rewriter optimized for faithfulness and overall quality, and an aesthetics rewriter tailored to visual appeal.

### F.2 PERFORMANCE FOR EACH ROUND

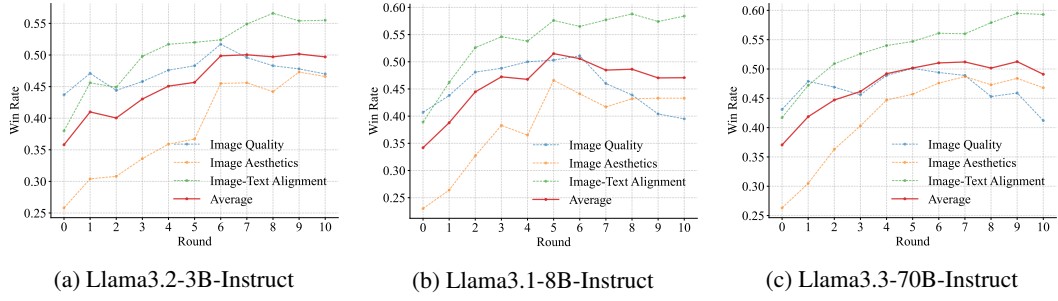

(a) Llama3.2-3B-Instruct          (b) Llama3.1-8B-Instruct          (c) Llama3.3-70B-Instruct

Figure 4: Win rate for different base models in different rounds. The prompt rewriters are evaluated on Pick-a-pic v2 with GPT-4o.

We report the win rates of different base models across DPO training rounds in Table 4. During the first six rounds, all metrics exhibit steady improvement. Beyond round six, however, image

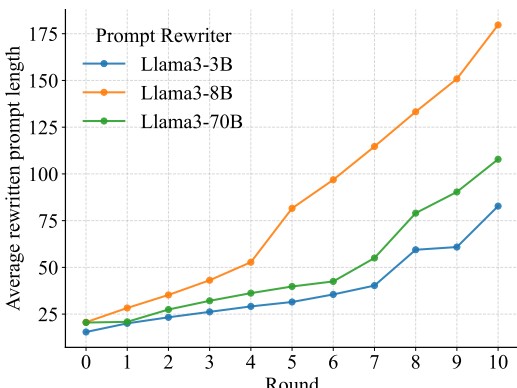

Figure 5: Average length of rewritten prompts across different DPO training rounds, computed on the Pick-a-Pic v2 dataset.

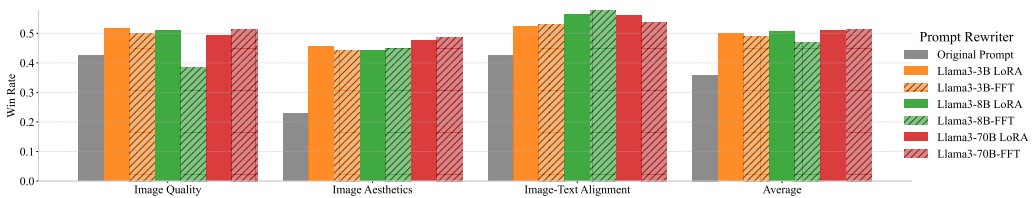

Figure 6: Full finetune vs. LoRA. Win rates against DALL-E 3. Evaluated on the Pick-a-pic v2 datasets with GPT-4o judge.

quality and aesthetics begin to decline, while text–image alignment continues to improve, reflecting an inherent trade-off. Consequently, the average win rate decreases, indicating that this trade-off becomes more pronounced after six rounds. It should also be noted that our training dataset contains approximately 60k samples, with 10k used per round; thus, the first six rounds effectively cover the entire dataset once. This observation suggests that expanding the dataset could yield further gains, and that a single iteration over the training data is sufficient, whereas additional iterations may lead to overfitting and degraded performance.

### F.3 Rewritten Prompt Length

Figure 5 illustrates a steady increase in the average token length of rewritten prompts over training. The rewriter progressively incorporates additional specificity, such as attributes, relations, and constraints, and this increase in length correlates with higher win rates.

### F.4 Full Finetune vs. LoRA

We conduct a controlled comparison between LoRA and full-model finetuning under identical experimental conditions, as shown in Figure 6. Full-model finetuning does not yield consistent improvements, whereas LoRA attains comparable performance while requiring substantially less memory and computation. These results suggest that LoRA constitutes a strong and practical choice for prompt-rewriter training.

## G Case Study

### G.1 General Rewriter

We provide five illustrative examples for the general rewriter in Tables 11, 12, 13, 14, and 15. In these examples, we employ Llama3-70B-Instruct as the prompt rewriter and FLUX.1-schnell as the backbone T2I model. To highlight the training dynamics, we present the rewritten prompts together

with their corresponding generated images across successive DPO training rounds, where round 0 corresponds to the original user-provided prompt. As training progresses, the rewritten prompts become increasingly detailed and precise, resulting in improvements in image quality, text–image alignment, and aesthetics.

## G.2 AESTHETICS REWRITER

We also observe that the aesthetics rewriter can, in some cases, compromise image–text alignment relative to the general rewriter, as illustrated by the examples in Table 16.

# H DETAILED REWARD DESIGN

In this section, we present the prompts employed for reward evaluation, corresponding to Figures 7, 8, 9, 10, and 11.

Table 11: Case study example 1.

| Round | Prompt | Image |
|-------|--------|-------|
| 0 | city skyline in visual novel |  |
| 1 | A vibrant cityscape serves as the backdrop in this visual novel, with a sprawling metropolis skyline featuring towering skyscrapers and bustling streets. |  |
| 2 | A vibrant cityscape serves as the backdrop for a dramatic scene in a visual novel, with towering skyscrapers and neon lights illuminating the night sky. |  |
| 3 | A futuristic cityscape serves as the backdrop for a dramatic scene in a visual novel, with sleek skyscrapers and neon lights illuminating the night sky as characters' dialogue and choices hang in the balance. |  |
| 4 | A futuristic cityscape at dusk, with sleek skyscrapers and neon lights illuminating the horizon, serves as the backdrop for a dramatic scene in a Japanese-style visual novel, complete with dialogue bubbles and character sprites. |  |
| 5 | A futuristic cityscape at dusk serves as the backdrop for a visual novel scene, with sleek skyscrapers and neon-lit billboards towering above, as characters in the foreground engage in a dramatic conversation amidst the bustling metropolis. |  |
| 6 | A futuristic cityscape at dusk serves as the backdrop for a visual novel interface, featuring a ornate user interface with anime-style characters and dialogue boxes amidst a sprawling metropolis with neon-lit skyscrapers, bustling streets, and a vibrant night sky. |  |

Table 12: Case study example 2.

| Round | Prompt | Image |
|-------|--------|-------|
| 0 | Movie poster artwork of a four-armed cyborg alien. |  |
| 1 | Vibrant movie poster featuring a futuristic, four-armed cyborg with an otherworldly alien physique. |  |
| 2 | Vibrant movie poster featuring a futuristic cyborg alien with four powerful arms, blending human and extraterrestrial elements in a stunning sci-fi design. |  |
| 3 | A futuristic movie poster featuring a powerful cyborg alien with four arms, surrounded by a vibrant, otherworldly landscape and explosive action elements. |  |
| 4 | A futuristic movie poster featuring a menacing four-armed cyborg alien with a metallic exoskeleton and glowing blue circuits, standing amidst a backdrop of exploding spaceships and a war-torn planetary landscape. |  |
| 5 | A sci-fi movie poster featuring a futuristic cyborg alien with four arms, surrounded by a cosmic landscape and advanced technology, with the title and credits of the film emblazoned in bold, neon-lit letters. |  |
| 6 | A sci-fi movie poster featuring a futuristic cyborg alien with four arms, adorned in metallic armor and surrounded by a cosmic landscape, with the title and credits of the film emblazoned in bold, neon-lit letters. |  |

Table 13: Case study example 3.

| Round | Prompt | Image |
|---|---|---|
| 0 | full body, a slender anime girl with long cyan hair, powerful arcane wizard, beautiful white outfit, extremely detailed, realistic shading |  |
| 1 | A stunning, slender anime girl with flowing cyan locks stands tall, radiating magical prowess as a powerful arcane wizard. She wears a breathtakingly beautiful white outfit, showcased in exquisite detail with remarkably realistic shading. |  |
| 2 | A stunning, slender anime girl stands tall, her lengthy cyan locks cascading down her back as she embodies the essence of a powerful arcane wizard. Adorned in a breathtakingly beautiful, intricately designed white outfit, she exudes magic and sophistication. The artwork boasts extremely detailed textures and realistic shading, bringing this enchanting scene to life. |  |
| 3 | A stunning anime sorceress stands tall, clad in an exquisite white ensemble, as her slender figure is surrounded by an aura of mystical energy. Her lengthy cyan locks cascade down her back like a river of ice, contrasting beautifully with the intricate, ornate details of her attire. With a commanding presence, she wields her magical prowess, as the atmosphere is electrified by her arcane abilities. The artwork boasts extremely realistic shading, bringing every facet of this enchanting scene to life with breathtaking precision. |  |
| 4 | A majestic, slender anime sorceress stands tall, adorned in a exquisite, intricately designed white attire, surrounded by a halo of mystical energy. Her long, flowing cyan hair cascades down her back like a river of celestial magic. With an ornate staff in hand, she channels powerful arcane forces, as evidenced by the swirling, ethereal auras and glowing runes that dance around her. The entire scene is rendered in breathtakingly detailed, hyper-realistic art, with masterful shading that imbues the image with depth, dimension, and an aura of wonder. |  |
| 5 | A majestic, slender anime sorceress stands tall, adorned in an exquisite, intricately designed white attire, surrounded by a halo of mystical energy. Her long, flowing cyan hair cascades down her back like a river of night sky, as she wields a ornate staff crackling with arcane power. Every aspect of her regal presence is rendered in breathtaking, hyper-realistic detail, from the delicate folds of her garments to the subtle, luminous glow of her magical aura, all set against a backdrop of subtle, gradient shading that imbues the entire scene with an aura of wonder and enchantment. |  |
| 6 | A majestic, slender anime sorceress stands amidst a mystical aura, adorned in an exquisite, intricately designed white attire adorned with golden accents and ornate gemstones, as she summons a vortex of magical energy with her staff. Her lengthy, vibrant cyan hair cascades down her back like a waterfall, with strands flowing upwards, surrounded by a halo of radiant, swirling runes. She poses amidst a lavish, ornate backdrop of ancient tomes, mystical artifacts, and celestial bodies, surrounded by a kaleidoscope of magical orbs, stars, and glittering, ethereal particles. The atmosphere is alive with dynamic, shimmering lights and intense, realistic shading, accentuating the intricate details of her elaborate costume, the ornate environment, and the fantastical, dreamlike scenery. |  |

Table 14: Case study example 4.

| Round | Prompt | Image |
|---|---|---|
| 0 | An stylized entrance to a rocky cave |  |
| 1 | A whimsical, artistic gateway to a rugged, rocky cavern. |  |
| 2 | A fantastical, ornate gateway leads to a mystical rocky cave, surrounded by ancient stones and lush greenery. |  |
| 3 | A fantastical, ornate gateway leading to a mystical rocky cave, surrounded by lush greenery and vines, with ancient ruins and mysterious artifacts scattered throughout the entrance. |  |
| 4 | A fantastical, ornate gateway adorned with mystical symbols and vines, leading to a majestic, ancient rocky cave surrounded by lush greenery and towering trees, with a warm, golden light emanating from within. |  |
| 5 | A fantastical, ornate gateway adorned with ancient carvings and mystical symbols, leading to a majestic, torch-lit rocky cave surrounded by towering stalactites and stalagmites, with a hint of misty aura and a starry night sky visible in the background. |  |
| 6 | A ornate, fantastical entrance adorned with ancient carvings and mystical symbols, leading to a majestic rocky cave surrounded by lush greenery and towering mountains in the background, with a warm glow emanating from within. |  |

Table 15: Case study example 5.

| Round | Prompt | Image |
|---|---|---|
| 0 | an anime girl in a kimono |  |
| 1 | A young woman with Japanese-inspired features and styling wears a traditional kimono, adorned with intricate designs and patterns, in a vibrant anime art style. |  |
| 2 | A beautiful anime girl adorned in a vibrant traditional Japanese kimono, surrounded by a serene atmosphere. |  |
| 3 | A beautiful anime girl adorned in a traditional Japanese kimono, surrounded by a serene atmosphere, with intricate designs and patterns on her garment, and a gentle smile on her face. |  |
| 4 | A beautiful anime girl adorned in a traditional Japanese kimono, surrounded by cherry blossoms, standing in a serene garden setting. |  |
| 5 | A beautiful anime girl adorned in a traditional Japanese kimono, surrounded by cherry blossoms, standing in a serene garden with a tranquil waterfall in the background. |  |
| 6 | A beautiful anime girl adorned in a traditional Japanese kimono, surrounded by cherry blossoms, posing elegantly in a serene Japanese garden. |  |

Table 16: Case study for the aesthetic rewriter. While it often enhances visual appeal, it can sometimes lead to prompt misalignment. For example, in Image 1, the result is more visually pleasing but rendered brighter, contradicting the user's low-light requirement. In Image 2, the rewriter overlooks the instruction to depict a fat man, producing a misaligned outcome.

| | No Rewriter | General Rewriter | Aesthetics Rewriter |
|---|---|---|---|
| Prompt1 | a low light photo of a city at night | A moody nighttime cityscape photograph captured in low light, depicting a bustling metropolis adorned with twinkling skyscrapers, neon signs, and streets aglow with vehicle headlights, set against a dark starry sky. | A breathtaking cityscape at dusk surrounded by a kaleidoscope of twinkling lights, majestic skyscrapers, and bustling streets, set against a backdrop of a vibrant purple and blue night sky filled with stars, captured in a stunning low-light photograph. |
| Image1 |  |  |  |
| Prompt2 | Fat black man in suit watching iguanas in a petting zoo realistic hyper realistic photorealistic | A photorealistic image depicting a obese African American man dressed in a formal business suit, standing in a lush petting zoo surrounded by various tropical plants and trees, intently observing a group of iguanas roaming freely around him, with some of the reptiles climbing on his legs and arms, as he gently offers them leaves to eat, amidst a backdrop of a sunny day with a clear blue sky. | A lavishly dressed black man surrounded by opulent decorations and lush greenery, sits in a luxurious golden chair, observing in wonder as a group of vibrant iguanas roam and play in a majestic petting zoo filled with exotic flowers, sparkling fountains, and intricate stone statues, surrounded by a backdrop of majestic waterfalls and a breathtaking sunset, amidst a scene of unparalleled realism, surrounded by intricate details and majestic architecture, adorned with gold ornaments and precious gems, surrounded by a warm and vibrant ambiance, illuminated by the golden light of the setting sun, in an incredibly detailed and intricate hyper-realistic artwork. |
| Image2 |  |  |  |

---

**Quality Reward**

You are an expert in analyzing images and observing any implausibilitys in the image. You will be provided with two images and you are tasked to compare them and decide which one is more plausible.
Here is a text caption for the two images, just for you to better understand the context: "{caption}".
Consider the following factors during your evaluation:

1. Human and Animal Body Parts:
-- Faces: Look for anomalies in facial features such as asymmetry, unnatural expressions, or distorted features.
-- Eyes and Gaze: Check for unnatural eye shapes, inconsistent gaze direction, or mismatched eye colors.
-- Teeth: Observe if the teeth appear odd, asymmetric, or have unnatural shapes and counts.
-- Ears and Earrings: Inspect for discrepancies in ear size, placement, or mismatched earrings.
-- Hands and Fingers: Count the fingers and examine the hands for unnatural poses or distortions.
-- Hair and Skin: Look for unrealistic hair strands, halos around the hair, or skin textures that lack pores and fine lines.

2. Geometry:
-- Straight Lines and Edges: Identify any wavy or misaligned lines that should be straight.
-- Perspective: Assess if the object's sizes and placements make sense within the scene.
-- Symmetry: Check for inconsistencies in symmetrical objects or features.
-- Relative Size: Ensure that objects are proportionally sized relative to each other.

3. Physics:
-- Reflections: Verify if reflections in mirrors, glasses, or water obey the laws of physics.
-- Shadows: Look for the presence and consistency of shadows based on the light source.
-- Objects Without Support: Identify any objects that appear to float without logical support.

4. Semantics and Logic:
-- Spatial Reasoning: Examine if objects are placed in a way that makes sense spatially.
-- Context and Scene Composition: Determine if the scene's elements logically fit together.
-- Other Semantic Issues: Look for impossible scenarios or illogical object interactions.

5. Text, Noise, and Details:
-- Text: Check for legible and correctly spelled text within the image.
-- Noise and Artifacts: Observe any unnatural noise patterns or color artifacts.
-- Fine-grained Details: Look for errors in detailed objects like clocks, keyboards, or fabrics.

6. Also, check for other kinds of implausibility not listed above.

You should analyze each image from the perspectives listed above, and make your output strictly adhere to the following format:
(Replace the <...> parts with your results. For parts outside the <> bracket, keep the exact words as they are part of the format. Note that: Images that look more plausible with little or no flaws are better. Images that contain more implausibilities are worse.)

Image Descriptions:

Image 1: <brief description>
Image 2: <brief description>

Image Quality Evaluation:

Image 1: <evaluation of image quality>
Image 2: <evaluation of image quality>
Image Quality Comparison: <A detailed description of your thorough analysis and reasoning for comparing Image 1 and Image 2 in image quality>

Final Image Quality Comparison Result: <One of the three outputs: Image 1 is better. Image 2 is better. It's a tie.>

Figure 7: Prompt for quality reward $r_{\text{Quality}}$.

## General Alignement Questions

You are an expert in understanding an image's caption, and asking comprehensive questions about the image to check if it aligns with the caption.

You will be provided with a caption. Try your best to come up with questions about the image such that by answering all the questions, one can determine whether the image depicts everything mentioned in the caption accurately.

Your output should only contain questions about the image based on the caption, and no other information. The questions should be listed line by line with bullet point symbols like "-". Here are some examples:

<Example Caption 1>:
"a zebra below a computer keyboard"

<Example Questions 1>:
- Does the image contain a zebra?
- Is there a computer keyboard in the image?
- Is the zebra positioned below the computer keyboard?
- Are both the zebra and the keyboard clearly visible and identifiable?
- Are the zebra and the keyboard the main focus of the image?

<Example Caption 2>:
"element"

<Example Questions 2>:
- "element" is a broad and abstract concept. Try to think of the many possible manifestations of "element". Does the image depict any angles of "element
- When looking at the image, does it remind you of the concept "element"?

<Example Caption 3>:
"David Bowie, very complex closeup macro portrait very complex hyper-maximalist overdetailed cinematic tribal fantasy closeup, shot in the photo studio, professional studio lighting, backlit, rim lighting, Deviant-art, hyper detailed illustration, 8k, symbolism Diesel punk, mist, ambient occlusion, volumetric lighting, Lord of the rings, BioShock, glamorous, emotional, tattoos,shot in the photo studio, professional studio lighting, backlit, rim lighting, Deviant-art, hyper detailed illustration, 8k"

<Example Questions 3>:
- Does the image feature David Bowie as the main subject?
- Is it a very complex close-up macro portrait of David Bowie?
- Does the portrait exhibit a hyper-maximalist and over-detailed style?
- Is there a cinematic and tribal fantasy theme present in the image?
- Was the portrait shot in a photo studio setting?
- Does the image utilize professional studio lighting techniques?
- Are there backlit and rim lighting effects applied?
- Is the style reminiscent of DeviantArt hyper-detailed illustrations?
- Is the image rendered in 8k resolution quality?
- Are elements of symbolism incorporated into the portrait?
- Does the image include Diesel punk aesthetics?
- Is there mist depicted within the scene?
- Are ambient occlusion and volumetric lighting techniques used?
- Does the image incorporate themes from "Lord of the Rings"?
- Are there visual influences from the game "BioShock"?
- Is the overall portrayal glamorous and emotional?
- Does David Bowie have tattoos in the image?
- Are the tattoos detailed and significant to the overall theme?
- Are all the stylistic elements cohesively integrated in the image?
- Is the composition focused on a hyper-detailed, cinematic presentation?

Now based on the examples, generate the questions for the target caption:

<Target Caption>:
"{caption}"

Your output should only contain questions about the image based on the caption, and no other information. The questions should be listed line by line with bullet point symbols like "-".

Figure 8: Prompt for general alignment questions.

> **General Alignement Reward**
>
> You are an expert in Visual Question Answering and checking the image's alignment with the caption. You will be provided with two images and a set of questions about the image. You are tasked to answer the questions for the two images and use that to guide your final assessment of the alignment of the given images with the caption.
> You should:
> 1) Briefly describe the images.
> 2) Answer the given list of questions for each image. The answer to each question can be "Yes", "Partially", or "No", then followed by an brief explanation (if needed).
> 3) Based on the answers to the questions, draw a conclusion about how the image aligns with the caption, and provide a comparison of the two images on which one is better.
>
> Note that: if the image depicts stuff that is not mentioned in the caption, it should NOT be considered as misalignment unless the additional stuff overwhelms the main focus.
>
> As a context, the caption used to describe the image is: "{caption}"
>
> The list of questions are:
> {question}
>
> Your output should be in this format:
>
> ==== Description of the image:
> Image 1: <brief description of image 1>
> Image 2: <brief description of image 2>
>
> ==== Answers to the questions:
> Image 1: <answers to the questions for image 1>
> Image 2: <answers to the questions for image 2>
>
> ==== Image-Text Alignment
> Image 1: <evaluation of image-text alignment for image 1>
> Image 2: <evaluation of image-text alignment for image 2>
> Image-Text Alignment Comparison: <A detailed description of your thorough analysis and reasoning for comparing Image 1 and Image 2 in image-text alignment>
>
> Final Image-Text Alignment Comparison Result: <One of the three outputs: Image 1 is better. Image 2 is better. It's a tie.>

Figure 9: Prompt for general alignment reward $r_{\text{General-Alignment}}$.

**Physical Alignement Reward**

You are an expert in checking if an image matches the provided text caption.
Carefully evaluate the caption and the two corresponding images based on image-text alignment. Text caption to be used for image-text alignment evaluation: "{caption}"
You should first briefly describe what are in the images. Then begin your assessment with a brief explanation that addresses the key factors listed. Following your explanation, provide a comparison of the two images on which one is better.

* Image-text alignment: note that for this criterion, you need to carefully review the image and the provided text caption, and check if the image depicts what the text caption describes. Better alignments corresponds to higher image-text alignment scores. The alignment can be further broken down into several aspects below:
1. Correct object generated and the number generated is correct. Pay attention to numbers in the text caption.
2. Good spatial relationship. Are objects generated with correct spatial relations? Pay attention to the caption's indication of spatial relation, such as key words like left, right, over, under, etc.
3. Correct attribute and attribute bindings. Are objects generated with the correct colors, shapes, and textures? Are the right attributes bound to the right objects?
4. Good world knowledge of history, geography, etc. Pay attention to names and check if image generated the right person or place, etc.
5. In general, is the image displaying good reasoning skill based on the text caption?

Note that: if the image depicts stuff that is not mentioned in the caption, it should NOT be considered as misalignment unless the additional stuff overwhelms the main focus.

Your output should strictly adhere to this format:
(Replace the <...> parts with your results. For parts outside the <> bracket, keep the exact words as they are part of the format.)

Image Descriptions:

Image 1: <brief description>
Image 2: <brief description>

Image-Text Alignment Evaluation:

Image 1: <evaluation of image-text alignment>
Image 2: <evaluation of image-text alignment>
Image-Text Alignment Comparison: <A detailed description of your thorough analysis and reasoning for comparing Image 1 and Image 2 in image-text alignment>

Final Image-Text Alignment Comparison Result: <One of the three outputs: Image 1 is better. Image 2 is better. It's a tie.>

Figure 10: Prompt for physical alignment reward $r_{\text{Physical-Alignment}}$

```
Aesthetics Reward

You are an expert in evaluating aesthetics of images.
Carefully evaluate the caption and the two corresponding images based on aesthetics. Here is the text caption for the two
images, just for you to better understand the context: "{caption}".

You should first briefly describe what are in the images. Then begin your assessment with a brief explanation that addresses the
key factors listed. Following your explanation, provide a comparison of the two images on which one is better.

* Aesthetics: basically you just tell me whether the image looks beautiful and pleasing. Does it look like a piece of art?

Your output should strictly adhere to the following format:
(Replace the <...> parts with your results. For parts outside the <> bracket, keep the exact words as they are part of the format.)

Image Descriptions:

Image 1: <brief description>
Image 2: <brief description>

Aesthetics Evaluation:

Image 1: <evaluation of aesthetics>
Image 2: <evaluation of aesthetics>
Aesthetics Comparison: <A detailed description of your thorough analysis and reasoning for comparing Image 1 and Image 2 in
aesthetics>

Final Aesthetics Comparison Result: <One of the three outputs: Image 1 is better. Image 2 is better. It's a tie.>
```

Figure 11: Prompt for aesthetics reward $r_{\text{Aesthetics}}$

