# OpenReview forum: "Improving Text-to-Image Generation with Input-Side Inference-Time Scaling"
_ICLR.cc/2026/Conference — ICLR 2026 Conference Withdrawn Submission_

### Official Review · Reviewer_jsjV · 2025-11-01

**Soundness:** 3
**Presentation:** 2
**Contribution:** 2
**Rating:** 2
**Confidence:** 5

**Summary:**

This paper addresses the limitation of text-to-image generation models in handling simple or underspecified prompts, which often result in suboptimal image-text alignment and visual quality. The authors propose a prompt rewriting framework that leverages large language models (LLMs) to refine user inputs before processing by T2I models. The approach employs a carefully designed reward system combined with iterative direct preference optimization (DPO) training, enabling the rewriter to enhance prompts without requiring supervised fine-tuning data.

**Strengths:**

1. The proposed approach trains a prompt rewriter using iterative direct preference optimization (DPO) with a multimodal LLM evaluator and a composite reward system covering image quality, aesthetics, and text-image alignment, requiring no supervised fine-tuning data.

2. Experimental results demonstrate that the framework significantly improves performance across multiple T2I models without modifying T2I backbones, exhibiting strong scalability and cross-model generalization capability.

**Weaknesses:**

1. Formulas 2 and 3 have formatting issues. References lack proper alignment with conference/journal standards. Overall writing quality and presentation (including typography, italics, and bold formatting) require improvement.
2. While experiments are conducted, the claims are not well-substantiated. A critical concern is that the MLLM judge significantly impacts results, yet there is insufficient discussion on ensuring reward confidence for each sample. The judgment capability of Qwen2.5-VL-7B appears limited and is not adequately validated. How is reward reliability ensured across diverse samples?
3. Key design choices and ablation studies on the reward system components are not clearly explained. The contribution of each reward component (image quality, aesthetics, text-image alignment) remains unclear.
4. How does the method maintain the rewriting capability for simple, well-specified prompts without over-modification? Is there risk of unnecessary rewriting degrading already good prompts?

**Questions:**

Have you considered joint optimization of the rewriter and judge simultaneously? Could end-to-end training improve overall performance and robustness?

---

### Official Review · Reviewer_qup4 · 2025-11-01

**Soundness:** 2
**Presentation:** 3
**Contribution:** 2
**Rating:** 4
**Confidence:** 4

**Summary:**

This paper proposes a prompt rewriting framework that improves text-to-image (T2I) generation by enhancing the input prompt rather than modifying the model. The method employs a large language model (LLM) rewriter trained through iterative Direct Preference Optimization (DPO) without supervised fine-tuning. For each user prompt, the rewriter generates several rewritten candidates, which are fed into a frozen T2I model to produce images. A multimodal LLM judge then evaluates these images on quality, aesthetics, and text–image alignment, producing pairwise preferences used to update the rewriter. Two variants are trained: a general rewriter (focused on text-image alignment and image quality) and an aesthetics rewriter (emphasizing visual appeal).

In experiments, the proposed method is tested on multiple T2I models (e.g., FLUX, Stable Diffusion 3.5, JanusPro) and benchmarks like Pick-a-Pic v2, GenEval, T2I-CompBench++, TIFA, and MS-COCO-30K. Results show consistent improvements in image–text alignment, fidelity, and aesthetics without retraining the underlying T2I models. Scaling experiments demonstrate that larger LLMs yield better performance, and rewriters trained on one T2I backbone transfer well to others, confirming strong generalization and scalability.

**Strengths:**

- This paper targets an input-side, inference-time pipeline that treats the T2I model as a black box and improves outputs solely by rewriting prompts, avoiding any backbone retraining/finetuning, which is simple, general, and practically appealing.
- Across multiple backbones (FLUX, SD-3.5, JanusPro) and benchmarks (GenEval, T2I-CompBench++, TIFA, COCO-30K), the proposed method consistently improves text-image alignment and image quality under a unified protocol.
- Results show scaling with LLM size and cross-T2I backbone transfer, supporting the claim that the rewriter generalizes beyond its training generator.

**Weaknesses:**

- The paper explicitly acknowledges RL-based prompt rewriting [r1], which adopts a similar approach with a smaller prompt rewriter and PPO instead of DPO for optimization, in Related Work. Another related work, self-rewarding VLM [r2], uses large vision-language models (VLMs) both as rewrit­ers and as reward models to optimize prompts via DPO as well. However, the experimental section in this paper does not benchmark against these most similar methods, but focusing on ICL and other systems instead, leaving the core claim “RL prompt rewriting without SFT is better” only partially substantiated. For a fair comparision, the authors should consider adapting the code for SFT/PPO rewriter [r1] and run it on the same LLM prompt rewriter and the same frozen T2I backbones, and report seed-averaged results. Please also consider benchmarking against self-rewarding VLM [r2].
- Training and evaluation rely heavily on MLLM reward models and judges (Qwen2.5-VL and GPT-4o), which makes cross-paper comparisons fragile if different reward models/judges are used.
- The proposed pipeline samples $n$ rewrites, renders $n$ images, and performs $O(n^2)$ pairwise judge calls per prompt—iterated across DPO rounds, which raises concerns about computational complexicity compared with simpler SFT/PPO approaches [r1].
- The method needs two separate rewriters, a General Rewrite and an Aesthetics Rewriter. Adding an aesthetics reward improves aesthetics but measurably hurts alignment, which complicates claims of across-the-board superiority versus single-policy alternatives.

[r1] Hao, Y., Chi, Z., Dong, L. and Wei, F., 2023. Optimizing prompts for text-to-image generation. Advances in Neural Information Processing Systems, 36, pp.66923-66939.
[r2] Hongji Yang, Yucheng Zhou, Wencheng Han, and Jianbing Shen. 2025. Self-Rewarding Large Vision-Language Models for Optimizing Prompts in Text-to-Image Generation. In Findings of the Association for Computational Linguistics: ACL 2025, pages 7332–7349, Vienna, Austria. Association for Computational Linguistics.

**Questions:**

- There are already many T2I reward models (like ImageReward [r3] and PickScore [r4]) and many VLM evaluator for T2I (like LLMScore [r5] and VQAScore [r6]). Why did the authors choose to implement their own reward VLM? Are there any advantages of their reward model against exsiting ones?
- Can the authors provide a side-by-side evaluation clarifying similarities and differences between their iterative DPO with external VLM judge and a self-rewarding VLM loop (where the VLM both proposes and scores) [r2]?

[r3] Xu, J., Liu, X., Wu, Y., Tong, Y., Li, Q., Ding, M., Tang, J. and Dong, Y., 2023. Imagereward: Learning and evaluating human preferences for text-to-image generation. Advances in Neural Information Processing Systems, 36, pp.15903-15935.
[r4] Kirstain, Y., Polyak, A., Singer, U., Matiana, S., Penna, J. and Levy, O., 2023. Pick-a-pic: An open dataset of user preferences for text-to-image generation. Advances in neural information processing systems, 36, pp.36652-36663.
[r5] Lu, Y., Yang, X., Li, X., Wang, X.E. and Wang, W.Y., 2023. Llmscore: Unveiling the power of large language models in text-to-image synthesis evaluation. Advances in neural information processing systems, 36, pp.23075-23093.
[r6] Lin, Z., Pathak, D., Li, B., Li, J., Xia, X., Neubig, G., Zhang, P. and Ramanan, D., 2024, September. Evaluating text-to-visual generation with image-to-text generation. In European Conference on Computer Vision (pp. 366-384). Cham: Springer Nature Switzerland.

---

### Official Review · Reviewer_FeGT · 2025-11-03

**Soundness:** 2
**Presentation:** 2
**Contribution:** 3
**Rating:** 4
**Confidence:** 4

**Summary:**

This paper studies how to improve text-to-image generation models, and proposes an input-side inference-time scaling framework. The framework uses large language models to refine user inputs for text-to-image generation. With a carefully designed rewarding system and an iterative direct preference optimization, the trained model rewrites the prompts with multiple iterations during training. Experimental results show that the proposed method can further improve the original text-to-image models on several benchmarks and outperforms several baselines on GenEval.

**Strengths:**

- Introducing inference-time scaling to text-to-image generation is an interesting topic and remains unexplored by related works.
- The paper conduct experiments on several benchmarks and demonstrates consistent gains upon the base text-to-image models.
- The experiments on scaling DPO training iterations demonstrate its training-time scalability.
- The composite reward function considers many dimensions of text-to-image generation, leading to a good optimization direction for the models.

**Weaknesses:**

- Potentially Misleading Terminology: The central framing of the method as "inference-time scaling" may be misleading. In common usage, "inference-time scaling" typically refers to techniques where the trained model can adaptively use more compute at the moment of inference to produce a better result (e.g., using more decoding steps or sampling more candidates). However, the scaling demonstrated in this paper occurs primarily during the training stage by training for more iterative DPO rounds. Once trained, the prompt rewriter itself seems to have a fixed computational cost at inference, which doesn't align with the conventional understanding of the term.
- Misaligned Experimental Focus and Presentation: The paper's presentation and experimental design do not fully support its core claim of "inference-time scaling." A significant portion of the main paper is dedicated to large benchmark comparison tables (Tables 1, 2, and 3), which, while valuable, showcase standard performance rather than exploring the scaling properties themselves. A more compelling experimental section would have focused on demonstrating how a user could leverage more compute with a trained rewriter at inference time.
- Incomplete Comparison with Key Baselines: The comparison to other state-of-the-art prompt rewriting methods is incomplete across the reported benchmarks. While strong baselines like PARM++ are included in the GenEval comparison (Table 4), they are notably absent from the evaluations on other crucial benchmarks. Besides, human evaluations are also necessary to prevent models hacking the metrics.

**Questions:**

**Questions**

- How does the model make use of additional inference-time compute after the iterative DPO training? Does the iterative training is conducted for each query?

**Suggestions**

- The presentation of Figure 3 could be improved for clarity. There is a noticeable blank space between the first bar and the second bar.

---

### Note · Authors · 2025-11-13

I have read and agree with the venue's withdrawal policy on behalf of myself and my co-authors.